



# Measurement of the ice-nucleating particle concentration with the Portable Ice Nucleation Experiment during the Pallas Cloud Experiment 2022

Alexander Böhmländer[1], Larissa Lacher[1], Romy Fösig[1], Nicole Büttner[1], Jens Nadolny[1], David Brus[2], Konstantinos-Matthaios Doulgeris[2], and Ottmar Möhler[1]

[1]Institute of Meteorology and Climate Research, Karlsruhe Institute of Technology, Karlsruhe, Germany
[2]Atmospheric Composition Research, Finnish Meteorological Institute, Helsinki, Finland

**Correspondence:** Ottmar Möhler (ottmar.moehler@kit.edu)

**Abstract.** The Portable Ice Nucleation Experiment (PINE) was deployed during the Pallas Cloud Experiment (PaCE) 2022 for a three-month-period at the Sammaltunturi station in autumn 2022. The station is located on top of a hill on the edge between sub-Arctic and boreal forest environments, typically receiving air masses from the Arctic and the south. Since clouds are frequently present at the station during autumn, the present aerosol particles can have a direct impact on cloud properties. Ice-nucleating particles (INPs) are aerosol particles that facilitate primary ice nucleation in supercooled cloud droplets. The PINE measured the INP concentration with a high temporal resolution of six minutes at different nucleation temperatures between 240 K and 252 K. The INP concentration varied exponentially with the freezing temperature and differences between different months were observed. The highest median INP concentration was measured during December over the whole temperature range, while during November the lowest median INP concentration was measured. The data presented here is useful to study aerosol-cloud interactions for a sub-Arctic location with minimal anthropogenic influence. The high temporal resolution allows to correlate the INP concentration with other measurements, such as size distribution data and meteorological data. In addition, the data provides the ice nucleating ability of arriving aerosol particles, which can be combined with models to study the nature, the source and the age of the INPs.

## 1 Introduction

Aerosol-cloud-interactions play a key role in the weather and climate on Earth (e.g., IPCC, 2021). Cloud condensation nuclei (CCN) lead to the nucleation of water droplets from water vapour, enabling cloud formation in the atmosphere (e.g., Tatzelt et al., 2022). Below $0\,°C$ liquid cloud droplets and ice crystals can co-exist (e.g., Pruppacher and Klett, 1997). Primary ice formation can only be initiated homogeneously, i.e. without a foreign substance, below about $-38\,°C$ (e.g., Koop et al., 2000). Mixed-phase clouds (MPCs) exist in the temperature range $-38 - 0\,°C$, where primary ice formation happens heterogeneously due to ice-nucleating particles (INPs). INPs are a rare subset of aerosol particles, which can be characterized by their nucleation temperature, i.e. at which temperature the aerosol particle acts as an INP (e.g., Vali et al., 2015). Even though their abundance is relatively small, INPs have a large impact on the phase of MPCs and with that on the radiative budget of clouds and their



precipitation (e.g., Kanji et al., 2017). Precipitation events are linked to the presence of an ice phase in clouds, especially at higher latitudes (e.g., Field and Heymsfield, 2015; Mülmenstädt et al., 2015; Heymsfield et al., 2020). The different radiative

properties of mixed-phase clouds have been investigated in relation to their phase in the literature (e.g., Bellouin et al., 2020; Storelvmo, 2017). Fan et al. (2017) studied orographic MPCs and found an increase in precipitation correlated to the INP concentration. In addition, Vergara-Temprado et al. (2018) found a strong link between clouds at the Southern Ocean and the INP concentration. The nature and sources of INPs are understudied, especially in remote regions as the sub-Arctic (e.g., Schmale et al., 2021). Kanji et al. (2017) report various sources of INP, with mineral dust being the dominant INP for nucleation

temperatures below about -15 °C (e.g., DeMott et al., 2015) and aerosols of biogenic origin (e.g., pollen, fungi, etc.) being already active as an INP close to 0 °C (e.g., Maki et al., 1974).

The present report details measurements with the Portable Ice Nucleation Experiment (PINE, Bilfinger SE, Möhler et al., 2021) during the Pallas Cloud Experiment (PaCE) 2022 (Brus et al., 2024). The PINE has been deployed in various field and lab studies prior and offers a high temporal resolution over a wide temperature range to study the INP concentration (Kunz

et al., 2022; Lacher et al., 2024; Brasseur et al., 2022; Vogel et al., 2024).

## 2 Observation site

The PINE was connected to a total heated inlet alongside other instrumentation at the Sammaltunturi station, which is part of the Pallas Atmosphere-Ecosystem Supersite in Finnish Lapland, hosted by the Finnish Meteorological Institute (FMI) (Asmi et al., 2021; Brus et al., 2024) and part of Global Atmosphere Watch (GAW), Integrated Carbon Observation System (ICOS),

European Monitoring and Evaluation Programme (EMEP) and the Aerosol, Clouds and Trace Gases Research Infrastructure (ACTRIS). The station is located on top of a hill about 170 km north of the Arctic Circle (67.9733° N, 24.1157° E, 565 m above sea level, Hatakka et al. (2003)). The tree line is approximately 100 m below the station. The vegetation around the station mainly consists of low vascular plants, lichen and moss (e.g., Lohila et al., 2015). The boreal forest below the tree line consists of pine, spruce and birch trees (e.g., Komppula et al., 2005). The anthropogenic impact on the aerosols at the station is

minor since it is located inside the Pallas-Yllästunturi National Park and far away from larger settlements (Lohila et al., 2015). Sammaltunturi station is an excellent location to monitor the background air composition in norther Europe for these reasons (e.g., Lohila et al., 2015). During autumn the station is around half of the time inside clouds, making the location ideal to study cloud droplets and crystals as well as their associated aerosol particles (Lohila et al., 2015). Air masses originating the Arctic are the dominant type throughout all seasons, albeit southern origins follow closely (Asmi et al., 2011).

## 50 3 Instrument operation

The PINE is a portable expansion-type cloud chamber, that is additionally temperature controlled. The PINE is fully automated and can be operatored for longer periods remotely. The measurement principle of PINE is detailed in the following paragraph (see also Möhler et al., 2021). During the PaCE-2022 campaign, the PINE was operated between 240 K and 252 K at a temporal

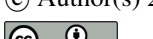



resolution of approximately 6 min. The air upstream of the PINE chamber is dried using two Nafion™ membrane dryers and its dew point temperature is measured directly before entering the chamber. The drying efficiency is controlled by the pressure of the drying air, which was controlled and monitored during the measurement period. The dew point temperature has to be slightly above the temperature that is reached during an expansion, but needs to be low enough that no frost forms on the walls of the chamber. Regular background measurements were performed, where the air stream was filtered using a HEPA filter (HEPA-CAP, Whatman™), to make sure frost does not lead to artefacts in the data. The PINE was remotely controlled using a custom-made LabVIEW Software.

The PINE operates on a fixed schedule, where three different modes cycle: *flush*, *expansion* and *refill*. The *flush* mode flushes the PINE chamber with ambient air containing the aerosol particles under investigation. This mode runs for $300\,\mathrm{s}$ at a flow of $2\,\mathrm{l_{std}min^{-1}}$. After the *flush* mode, the *expansion* mode starts. The chamber is evacuated to $790\,\mathrm{hPa}$ at a flow of $3\,\mathrm{l\,min^{-1}}$, the outflowing air passes the optical particle counter (OPC, fidas-pine, Palas GmbH), which measures the optical size of all aerosol particles above a certain size. This lower limit is dependent on the photo multiplier voltage of the detector. A volumetric flow is used for the expansion to keep the velocity of the air constant through the OPC. The trigger voltage of the OPC is adjusted to detect larger liquid cloud droplets as well as the ice crystals formed if INPs are present, according to the different ambient conditions (i.e. temperature and relative humidity). Ice crystals appear larger than liquid droplets in the OPC data, therefore the liquid and solid phases are separated by size. After reaching the set pressure, the chamber is refilled with ambient air during the *refill* mode at a flow of $2\,\mathrm{l_{std}min^{-1}}$. The inlet flow is always kept constant at $2\,\mathrm{l_{std}min^{-1}}$.

## 4 Data evaluation and quality control

The data of PINE is structured into two data levels: raw data (`raw_Data`) and level 1 data (`L1_data`). The first stage contains the raw data from the housekeeping data, i.e. temperature and pressure data as well as the single particle data of the OPC. The `L1_data` is created by the custom-made Python software PINE INP Analysis (PIA) v2.0.2 (Büttner et al., 2024, in preparation). The software uses the SaQC Python module to introduce flags to the dataset (Schäfer et al., 2024). The automatic flags are structured into "info", "warning" and "error" flags. The dataset is carefully evaluated and manually checked. In cases where the automatic flagging was not able to identify an error in the measurement, a manual flag was added.

One critical part of the analysis with the PIA software is the automatic ice threshold finder. The ice crystals are separated from the liquid cloud via their size information. The algorithm calculates the binned size distribution and searches for a minima in the data. This minima is considered the ice threshold above which only ice crystals are optically detected. In rare cases, this minima is not found and manually selected by following the algorithm:

1. A faulty threshold is identified for a given run.

2. Earlier thresholds are observed and if not faulty, the temporally closest threshold is selected.

3. As in 2, later thresholds are observed and if not faulty, the temporally closest threshold is selected.

4. The higher threshold of the two surrounding ones is selected as the new manual threshold.



**Table 1.** Additional variables contained in the associated dataset and their description.

| Variable | Description |
| --- | --- |
| `lon` | longitude |
| `lat` | latitude |
| `alt` | altitude above mean sea level |
| `meas_height` | inlet height above mean sea level |

This algorithm leads to a conservative estimate of the INP concentration with a possible low bias.

## 5 Overview of dataset

The dataset is given as a NetCDF file following the CF-1.11 metadata conventions. It contains the `L1_data` with the temporal resolution given by the instrument. The minimal temperature reached during an expansion is given in the variable `T_min`
(see also discussion in Möhler et al., 2021). The INP concentration associated with this temperature is given in the variable `INP_cn`. In addition, the dataset also contains the lower and upper bin boundary for the temperature binned data: `temp_low` and `temp_upp`, respectively.

Flags are added under the ancillary variable `INP_cn_qc` which contains the information on the "warning" and "error" flags provided by the PIA software as well as the manual flags set. Data is considered as invalid if the associated ancillary variable `INP_cn_qc` contains an "error" and/or a "manual" flag. The invalid data is not removed from the dataset. Additional variables and their description are listed in Table 1.

The INP concentration as a time series is shown as a coloured scatter plot in Figure 1. The colour is associated with the temperature $T$ (`T_min`) shown on the colour bar. Invalid data is removed in the visualization. Data where the temperature was above $T = 252\,\mathrm{K}$ was also removed for the visualisation (7 expansions). Zeros are not shown in the logarithmic plot, but they amount to around $35\,\%$ of the total number of expansions. Zeros happen when the INP concentration is below the sensitivity of PINE, which is around $0.5\,\mathrm{l_{std}^{-1}}$, which is especially relevant for higher temperatures ($\gtrsim 247\,\mathrm{K}$). Between 2022-09-23 and 2022-12-23 around 19600 valid expansions where performed in the temperature range $T = 239.83\,\mathrm{K}$ to $T = 266.34\,\mathrm{K}$. The temperature distribution is shown as a bar plot in Figure 2. The highest frequency of measurements is in the half-open temperature bin $T = (247, 249]\,\mathrm{K}$. The highest concentration of $c_{\mathrm{INP}} = 108.495\,\mathrm{l_{std}^{-1}}$ was measured on 2022-11-10 07:59:39+0000 at a temperature of $T = 241.67\,\mathrm{K}$. Large short-term fluctuations of the INP concentration are observed, which would not be visible with filter-based methods, that perform an integral measurement over longer time scales. The INP concentration split into the four months is shown in Figure 3. The temperature $T$ is the middle temperature of the temperature bins as specified in the previous paragraph. In general, the INP concentration is highest during December, while the lowest concentration is seen during November. The median of the four months visualized show large differences, sometimes deviating almost one order of magnitude (see $T = 242\,\mathrm{K}$ and October / November). The overall INP concentration increases exponentially with decreasing

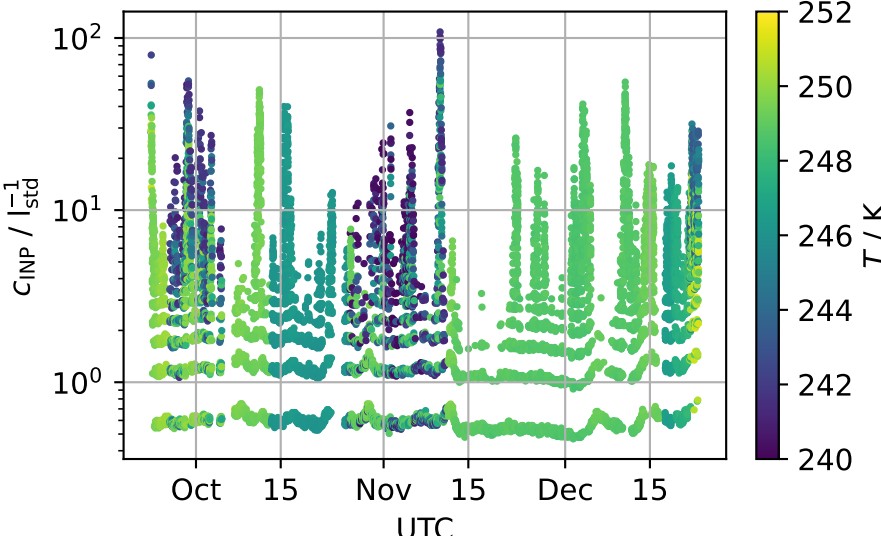

**Figure 1.** The INP concentration measured with PINE during the PaCE-2022 campaign as a timeseries. Each scatter point represents one expansion and its colour is associated with the minimal temperature $T$ during the expansion.

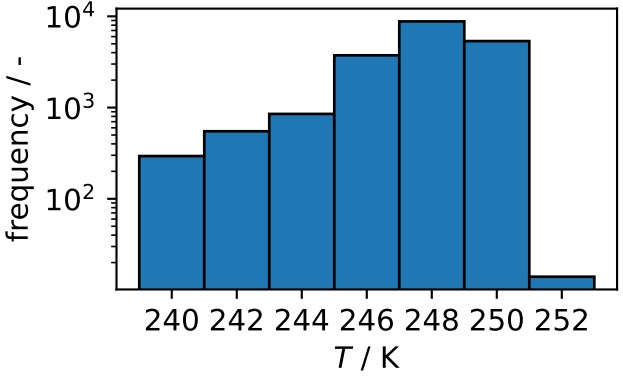

**Figure 2.** The frequency distribution of the minimal temperatures reached by the PINE during expansions during the PaCE-2022 campaign.

temperature. Especially at higher temperatures (i.e. > 246 K) the INP concentration is close to the sensitivity limit of PINE for all months except December.

## 6 Conclusions

The dataset provided is unique for the Sammaltunturi station. Rapid changes are visible in the dataset, that are not visible when using filter-based methods. Additional datasets are available that provide information on the type of aerosol particles,

115



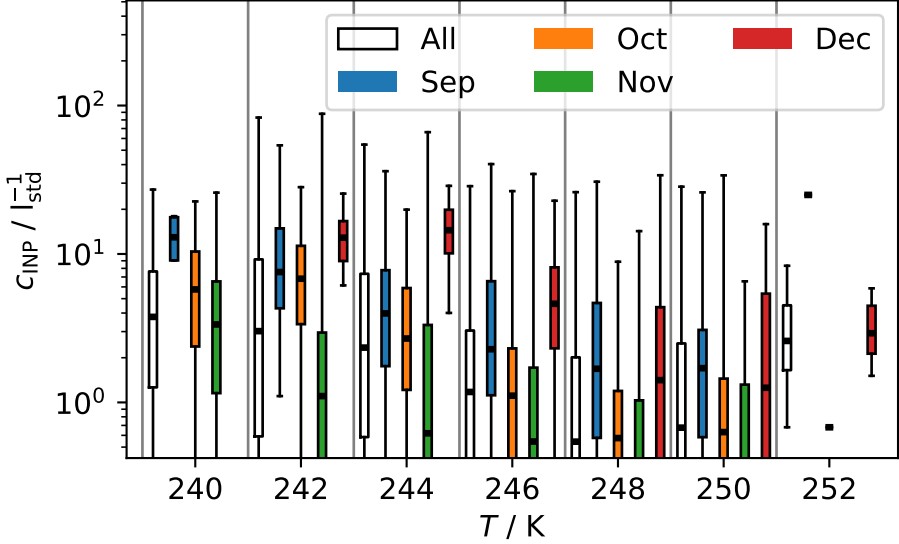

**Figure 3.** Boxplot representation of the INP concentration measured by the PINE during the PaCE-2022 campaign. The whiskers show the (1, 99) percentiles and the fliers are not visible, otherwise standard boxplot notation is used. Medians that are not visible are zero and therefore not visualized on the logarithmic scale.

i.e. fluorescent aerosol particles (Gratzl et al., 2025), the size and concentration of aerosol particles and meteorological data (Backman et al., 2025). This enables the use of the data to connect to small-scale modelling, for example using the flexible particle dispersion model (FLEXPART) to estimate the source and age of aerosols (e.g., Pisso et al., 2019). Aerosol data is available as well, providing aerosol concentration and size distribution data. The PINE also adds a source of reference for other data taken during the PaCE-2022 campaign utilizing uncrewed aerial vehicles (UAVs) and balloon-borne setups. The data is provided in its highest temporal resolution, allowing the data to be resampled to a time frame that is relevant for the investigation. In the grand scheme of the PaCE-2022 campaign various interesting cases were observed, which will result in a better understanding of the sources and impact of aerosols, especially in relation to aerosol-cloud interactions.

## 7 Code and data availability

Datasets are archived under individual DOI at the Zenodo Open Science data archive (https://doi.org/10.5281/zenodo.13889647, last access: 28022025, Böhmländer et al. (2025)), where a dedicated community Pallas Cloud Experiment - PaCE2022 has been established (https://zenodo.org/communities/pace2022/, last access: 28022025). This community houses the data files along with additional metadata on the datasets. The PIA software is available on a public gitlab instance under https://codebase.helmholtz.cloud/pine/pia_software. The SaQC software used for the flagging of the data is published under https://doi.org/10.5281/zenodo.5888547 (Schäfer et al., 2024).



*Author contributions.* AB did the data analysis and wrote the manuscript. LL and AB performed the measurements with the PINE during the PaCE-2022 campaign. RF and NB are responsible for the software development of the PIA software. JN developed the LabVIEW software to control and interact with the PINE instrument. DB and KD prepared and organized the PaCE-2022 campaign. All authors contributed to the proof reading and discussion of the dataset.

135 *Competing interests.* The contact author has declared that none of the authors has any competing interestes.

*Acknowledgements.* This research has been supported by the ACTRIS IMP GA 871115, the ACTRIS-Finland funding through the Ministry of Transport and Communications, and the Atmosphere and Climate Competence Center Flagship funding by the Research Council of Finland (Grants 337552).

The KIT project contribution was supported by the Helmholtz Association through the research program "Changing Earth - Sustaining our
140 Future". The authors would like to thank the technical team at the Sammaltunturi station for their support during the campaign, and the PINE team at KIT for continuous support in developing and operating the PINE instruments.



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
