# Peer review of "Measurement of the ice-nucleating particle concentration with the Portable Ice Nucleation Experiment during the Pallas Cloud Experiment 2022"

_Earth System Science Data, 2025_

## Author Comment (AC1)

**Author response to RC1**

August 26, 2025

We are grateful for the helpful comments and suggestions from the reviewer. Below the reviewers' comments are in blue with our responses in black directly below.

I have read the manuscript by Dr Böhmländer et al. and checked the associated dataset. In my opinion, this dataset is extremely valuable for the atmospheric scientific community, considered the scarcity and relevance of INP data. Additionally, INP concentration data are provided at high time resolution which is certainly an added value for investigating INP properties, sources and impact. Nevertheless, before publication some issues should be addressed in the manuscript.

My main concern regards the measurement uncertainty. This is a fundamental aspect characterising the dataset, yet it is not discussed at all in the manuscript, nor uncertainty ranges are presented in the data files. I would invite the authors to address this aspect of the data quality in the text and, if possible, to associate uncertainty "bars" to the provided data values.

One of the strong points of the dataset is the high temporal resolution, which provides valuable information on the short-term variability of INP concentration at the study site. I would invite the authors to provide some further quantitative information about this in describing the dataset. I understand that the main topic of a data journal is not the scientific exploitation of the data, but some further information can be provided. For instance, which is the extent of the short-term variability in INP concentration? How does it compare with the day to day and seasonal variability? Is the short-term variability constant trough the study periods or does it vary with season?

We thank the reviewer for their comments and we added a separate section in the manuscript to discuss the measurement uncertainty according to the "Evaluation of measurement data" (*Evaluation of measurement data - Guide to the expression of uncertainty in measurement* 2008) (lines 129-147).

old

new The uncertainty budget of PINE has a type B uncertainty (according to *Evaluation of measurement data - Guide to the expression of uncertainty in measurement* 2008) related to the individual uncertainties of the expansion duration $t$, the expansion flow $F$ and the ice crystal number $N_{\mathrm{INP}}$ measured by the OPC

$$c_{\mathrm{INP}} = \frac{N_{\mathrm{INP}}}{F \times t} \; . \tag{1}$$

The uncertainty of the expansion flow is given as

$$\sigma_F = 0.5\,\%\mathrm{RD} + 0.1\,\%\mathrm{FS} \; , \tag{2}$$

where RD and FS denote the reading value and the full scale of the mass flow controller (MFC, EL-FLOW Select F-201CV, Bronkhorst High-Tech B.V.), respectively. The largest value, considering an expansion flow of $3\,\mathrm{l_{std}min^{-1}}$, is therefore $0.025\,\mathrm{l_{std}min^{-1}}$, which corresponds to a relative uncertainty for the flow of approximately $0.83\,\%$, which is considered to be negligible. The uncertainty of the expansion duration just depends on the response time of the solenoid valve (Solenoid Valve XLS-16, SMC Corporation), which is below $0.2\,\mathrm{s}$, and therefore is also considered to be negligible. The uncertainty of the ice crystal number depends on the OPC (fidas-pine, Palas GmbH). In the prototype version of PINE, a different OPC was used (welas, Palas GmbH, see Möhler et al. 2021). The previous OPC used a T-shaped optical detection volume to detect particles and had an uncertainty of $20\,\%$ (Benz et al. 2005). Since the new OPC measures all particles and not just a small percentage of the total particle number, it can be assumed that the uncertainty of the new OPC (fidas-pine) is smaller. We assume a conservative estimate of $10\,\%$ for the relative uncertainty of

the ice crystal number. In addition, the type A uncertainty is related to a Poisson distribution for the ice crystal number. Combined, this leads to a total uncertainty for the INP concentration of

$$c_{\text{INP}} = c_{\text{INP}}^{\text{measured}} \pm 0.1 \times c_{\text{INP}}^{\text{measured}} \pm \frac{c_{\text{INP}}^{\text{measured}}}{\sqrt{N_{\text{INP}}}} \ , \tag{3}$$

where the first term is a type B uncertainty and the second term the type A uncertainty.
* * *
Regarding the second remark, we have added an additional statement looking at the interquartile range of the hourly INP concentration (lines 120-123):

old

new Looking at hourly INP concentration data measured in the half-open temperature bin $T = (247, 249]$ K, the mean of the interquartile range is $c_{\text{INP}} = 0.9\,\text{L}_{\text{std}}^{-1}$ with a maximum of the interquartile range of $c_{\text{INP}} = 21.6\,\text{L}_{\text{std}}^{-1}$. The interquartile range can be used to assess the spread of data.

We do not see any clear seasonal variability, but we will investigate this in more depth in a future publication, also taking into account additional data collected during PaCE-2022.

**1 Specific comments**

L12. It is not clear what the authors mean with "arriving aerosol particles"; are they referring to long range transport, maybe? Please reformulate this sentence for major clarity.

We mean aerosol particles, that enter the inlet of the measurements station, to make this more clear we have adjusted this sentence (lines 12-13).

old In addition, the data provides the ice nucleating ability of arriving aerosol particles, which can be combined with models to study the nature, the source and the age of the INPs.

new In addition, the data provides the ice nucleating ability of ambient aerosol particles, which can be combined with models to study the nature, the source and the age of the INPs.
* * *
L104. cINP = 108.495 L−1: I am wondering if all the figures are significant in this concentration.

We agree and therefore have adjusted this to the smallest resolution of PINE, which is around $0.5\,\text{L}_{\text{std}}^{-1}$ (line 118).

old The highest concentration of $c_{\text{INP}} = 108.495\,\text{L}_{\text{std}}^{-1}$ was measured on 2022-11-10 07:59:39+0000 at a temperature of $T = 241.67$ K.

new The highest concentration of $c_{\text{INP}} = 108.5\,\text{L}_{\text{std}}^{-1}$ was measured on 2022-11-10 07:59:39+0000 at a temperature of $T = 241.67$ K.
* * *
I have noticed some discrepancies between the data files and the manuscript. (1) the "flag" data column is labelled "INP_qc" in the data file, while it is "INP_cn_qc" in the text and in the data info file. (2) The column "INP_cn_flush" is "INP_flush" in the data info file. (3) There is not information on the meaning of the "INP_flush" data column in the data info file.

We appreciate the careful checking of the data variables and have added a paragraph to describe the variable of "INP_cn_flush" (lines 103-106):

old

new The "INP" concentration measured during the flush mode is given in the variable `INP_cn_flush`. This assumes that all larger aerosols or ice crystals from the walls that are detected during the flush mode above the ice threshold, might appear as an INP during the expansion. This value is typically much lower than the measured INP concentration (`INP_cn`) and is flagged as a "warning" if the condition `INP_cn_flush` $> 0.5 \times$ `INP_cn` is true.

We are unsure what the reviewer means regarding the naming of the `INP_qc` data variable, since the data variables are named as `INP_cn_qc` in the NetCDF files and also in the `INP_cn` under the ancillary variable attribute.
* * *
**References**

Benz, S. et al. (2005). "T-dependent rate measurements of homogeneous ice nucleation in cloud droplets using a large atmospheric simulation chamber". In: *Journal of Photochemistry and Photobiology A: Chemistry* 176.1–3, pp. 208–217. ISSN: 1010-6030. DOI: `10.1016/j.jphotochem.2005.08.026`.

*Evaluation of measurement data - Guide to the expression of uncertainty in measurement* (2008). Working Group 1 of the Joint Committee for Guides in Metrology.

Möhler, O. et al. (2021). "The Portable Ice Nucleation Experiment (PINE): a new online instrument for laboratory studies and automated long-term field observations of ice-nucleating particles". In: *Atmospheric Measurement Techniques* 14.2, pp. 1143–1166. DOI: `10.5194/amt-14-1143-2021`.

---

## Author Comment (AC2)

**Author response to RC2**

August 26, 2025

We are grateful for the helpful comments and suggestions from the reviewer. Below the reviewers' comments are in blue with our responses in black directly below.

This paper describes a unique dataset of INP concentrations taken using a cloud chamber-style INP measurer, the Portable Ice Nucleation Experiment. The unusually high temporal resolution of INP measurements offers rare opportunities for future users to investigate correlations between INP concentrations and other high-temporal resolution data.

The paper is mostly well-written and the data is well-presented and highly accessible. Following best practice, the authors have not removed data that should be excluded based on quality control flags (which are provided), and I was able to replicate the basic plots in Figs. 1 and 2 within a few minutes of downloading the dataset using basic plotting tools in Python, which is commendable.

However, I have some minor concerns regarding the measurement uncertainty, and one undocumented piece of data contained in the files. I also think this paper occasionally relies on the reader having experience of INP measurements, which is not necessarily the case for some readers of this journal given its interdisciplinary nature.

If these minor concerns are addressed, I believe this paper will be an excellent contribution to ESSD.
* * *
**1 Specific comments on the paper**

Throughout: The authors point to other papers which discuss uncertainties associated with PINE, but the review criteria in ESSD requires that "error estimates and sources of errors [are] given (and discussed in the article)". There is a brief statement that concentrations may be biased low (line 86) but there is no indication of how low. It would be helpful for the authors to add a short discussion of the uncertainty here in order that users of this data can easily consider this.

We added a new section to discuss the uncertainties of the measurements (see also answer to reviewer 1, lines 129-147).

old

new The uncertainty budget of PINE has a type B uncertainty (according to *Evaluation of measurement data - Guide to the expression of uncertainty in measurement* 2008) related to the individual uncertainties of the expansion duration $t$, the expansion flow $F$ and the ice crystal number $N_{\mathrm{INP}}$ measured by the OPC

$$c_{\mathrm{INP}} = \frac{N_{\mathrm{INP}}}{F \times t} \ . \tag{1}$$

The uncertainty of the expansion flow is given as

$$\sigma_F = 0.5\,\%\mathrm{RD} + 0.1\,\%\mathrm{FS} \ , \tag{2}$$

where RD and FS denote the reading value and the full scale of the mass flow controller (MFC, EL-FLOW Select F-201CV, Bronkhorst High-Tech B.V.), respectively. The largest value, considering an expansion flow of $3\,\mathrm{l_{std}min^{-1}}$, is therefore $0.025\,\mathrm{l_{std}min^{-1}}$, which corresponds to a relative uncertainty for the flow of approximately $0.83\,\%$, which is considered to be negligible. The uncertainty of the expansion duration just depends on the response time of the solenoid valve (Solenoid Valve XLS-16, SMC Corporation), which is

below 0.2 s, and therefore is also considered to be negligible. The uncertainty of the ice crystal number depends on the OPC (fidas-pine, Palas GmbH). In the prototype version of PINE, a different OPC was used (welas, Palas GmbH, see Möhler et al. 2021). The previous OPC used a T-shaped optical detection volume to detect particles and had an uncertainty of 20 % (Benz et al. 2005). Since the new OPC measures all particles and not just a small percentage of the total particle number, it can be assumed that the uncertainty of the new OPC (fidas-pine) is smaller. We assume a conservative estimate of 10 % for the relative uncertainty of the ice crystal number. In addition, the type A uncertainty is related to a Poisson distribution for the ice crystal number. Combined, this leads to a total uncertainty for the INP concentration of

$$c_{\mathrm{INP}} = c_{\mathrm{INP}}^{\mathrm{measured}} \pm 0.1 \times c_{\mathrm{INP}}^{\mathrm{measured}} \pm \frac{c_{\mathrm{INP}}^{\mathrm{measured}}}{\sqrt{N_{\mathrm{INP}}}} \ , \tag{3}$$

where the first term is a type B uncertainty and the second term the type A uncertainty.

We also clarified our statement in line 86 to reflect that this low bias might originate from setting a faulty threshold manually to the higher ice threshold. A higher ice threshold means that there are potentially ice crystals that are not classified as an INP (lines 96-97).

old This algorithm leads to a conservative estimate of the INP concentration with a possible low bias.

new This algorithm leads to a conservative estimate of the INP concentration with a possible low bias due to selecting the higher threshold in step 4.
* * *
Throughout: The authors largely attribute the importance of this dataset to the high temporal resolution and highlight on line 114 that "rapid changes are visible in the dataset". Could the authors: - demonstrate that the dataset is able to capture "short-term fluctuations in INP concentration" (line 105)? It may be helpful to do this by showing a subset of a small time-period of data alongside the whole campaign's data depicted in Figure 1. - briefly describe the current dearth of high-temporal resolution measurements in the introduction (see also comment 8)?

We have added a short discussion on the extend of short-term changes of the INP concentration (see also answer to reviewer 1, lines 120-123):

old

new Looking at hourly INP concentration data measured in the half-open temperature bin $T = (247, 249]$ K, the mean of the interquartile range is $c_{\mathrm{INP}} = 0.9\,\mathrm{L}_{\mathrm{std}}^{-1}$ with a maximum of the interquartile range of $c_{\mathrm{INP}} = 21.6\,\mathrm{L}_{\mathrm{std}}^{-1}$. The interquartile range can be used to assess the spread of data.

We also added a sentence to the introduction to discuss high temporal measurements in the Arctic and sub-Arctic (see also answer to comment 8, lines 33-36):

old

new Previous measurements of the INP concentration in the boreal forest showed a dominant contribution from local biogenic aerosol to the total INP population (Schneider et al. 2020; Vogel et al. 2024), however, given the measurements location in the sub-Arctic, the INP population might be different from the one studied in Hyytiälä. The vast majority of INP concentration measurements in the sub-Arctic and Arctic are filter-based with a low temporal resolution of typically several days (e.g., Tobo et al. 2024; Wex et al. 2019).
* * *
Lines 24-28: I found this section to be a little unclear at times. To make the link between INP and cloud phase more explicit, when discussing the Vergara-Temprado (2018) paper, could the authors point out that INP were shown to control cloud reflectivity?

We have clarified our statement regarding the influence of the INP concentration on clouds in the Southern Ocean as shown by Vergara-Temprado et al. 2018 (lines 27-28):

old In addition, Vergara-Temprado et al. (2018) found a strong link between clouds at the Southern Ocean and the INP concentration.

new In addition, Vergara-Temprado et al. (2018) showed that the INP concentration in the Southern Ocean controls the reflectivity of MPCs.
* * *
Line 31 and Section 2: If a particular INP species is known to dominate at Pallas, could the authors consider specifying which INP types are targeted by these measurements?

Generally, it is not known specifically for Pallas, which INP species dominate. From previous measurements in the boreal forest (i.e., Hyytiälä Schneider et al. 2020), it was shown, that snow cover leads to a reduction of the INP concentration at temperatures between around 250 K and 265 K. Furthermore, a strong seasonal cycle was visible in the INP concentration, and together this was linked to an abundance of biogenic aerosol. Furthermore, Vogel et al. 2024 also found a moderate correlation between the INP concentration measured below around 249 K and the fluorescent bioaerosol concentration.

During PaCE-2022, a Wideband Integrated Bioaerosol Sensor (WIBS) was deployed alongside PINE and a filter-based INP technique, together with other instrumentation to investigate biogenic aerosols and their impact on the INP population at Pallas. Gratzl et al. 2025 found a similar behaviour as in Hyytiälä, with a strong seasonal trend (between September 2022 and September 2023) and a reduction with snow cover of highly fluorescent aerosol particles (HFAPs). The HFAP concentration has a positive correlation with the INP concentration active between around 242 K and 265 K, with the highest correlation coefficient seen for INPs above 259.65 K ($r = 0.94$, $p < 0.001$).

While biogenic aerosols definitely are one of the defining features of biogenic INPs in the boreal forest, we are also interested in mineral dust via long-range transport. This is especially interesting with regard to high-latitude dust, which might play a more dominant role than previously thought in the sub-Arctic and Arctic (e.g., Dagsson-Waldhauserova et al. 2017; Groot Zwaaftink et al. 2017; Meinander et al. 2025).

We have added a short summary of this to the manuscript and plan to publish a paper discussing the source of the measured INPs in more detail, specifically regarding the PaCE-2022 campaign (lines 31-33):

old

new Previous measurements of the INP concentration in the boreal forest showed a dominant contribution from local biogenic aerosol to the total INP population (Schneider et al. 2020; Vogel et al. 2024), however, given the measurements location in the sub-Arctic, the INP population might be different from the one studied in Hyytiälä. The vast majority of INP concentration measurements in the sub-Arctic and Arctic are filter-based with a low temporal resolution of typically several days (e.g., Tobo et al. 2024; Wex et al. 2019).
* * *
Lines 48-49: Table 2 of the Asmi paper cited says that between Nov and Feb, the Arctic and Southern origin airmasses have frequencies of 31 and 28% respectively, suggesting that the Arctic air is not "dominant". Additionally, the Asmi definition of the Arctic is a few degrees above the Arctic Circle. Could the authors describe the airmass origins with more precision?

We agree with the reviewer, that our statement was not accurate, we have therefore adjusted it (lines 51-54):

old Air masses originating the Arctic are the dominant type throughout all seasons, albeit southern origins follow closely (Asmi et al. 2011).

new Air masses typically originate in the Arctic during autumn (Sep-Oct) and winter (Nov-Feb) with a frequency of 36 % and 31 %, respectively (Asmi et al. 2011). Southern air masses follow closely with frequencies of 26 % and 28 % during autumn and winter, respectively (Asmi et al. 2011).

We are currently planning on performing FLEXPART (FLEXible PARTicle dispersion model, Pisso et al. 2019) simulations for the PaCE-2022 campaign, but currently we are unable to give a precise answer for the air mass origins during the campaign.
* * *
Line 65: In this setup, what is the "certain size" below which aerosols are not measured?

The certain size does depend on the photo multiplier voltage used during the measurements. Typically, the certain size is around $2\,\mu m$, but we do not calibrate the fidas-pine sensor prior to each campaign. The sensor has to be able to observe the liquid cloud droplets as well as the larger ice crystals and allow for a clear separation via size. We have added this information to the manuscript (lines 67-69):

old The chamber is evacuated to $790\,hPa$ at a flow of $3\,lmin^{-1}$, the outflowing air passes the optical particle counter (OPC, fidas-pine, Palas GmbH), which measures the optical size of all aerosol particles above a certain size.

new The chamber is evacuated to $790\,hPa$ at a flow of $3\,lmin^{-1}$, the outflowing air passes the optical particle counter (OPC, fidas-pine, Palas GmbH), which measures the optical size of all aerosol particles above a certain size (around $2\,\mu m$).
* * *
Lines 67-68: Could the authors explicitly describe how the INP concentration is counted; i.e. the number of ice crystals is determined to be the number of immersion-mode INP (Möhler, et al. 2021)?

We added the information regarding immersion freezing (lines 70-72):

old The trigger voltage of the OPC is adjusted to detect larger liquid cloud droplets as well as the ice crystals formed if INPs are present, according to the different ambient conditions (i.e. temperature and relative humidity).

new The trigger voltage of the OPC is adjusted to detect larger liquid cloud droplets as well as the ice crystals formed via immersion freezing and deposition nucleation if INPs are present, according to the different ambient conditions (i.e. temperature and relative humidity). The immersion freezing is the dominant mode, but, especially at the start of the expansion, ice crystals might also be formed via deposition nucleation.
* * *
Line 115: In the vein of comment (2.2), while readers familiar with INP measurements will be aware of the challenges of measuring INP at high temporal resolution, consider pointing readers less familiar to papers which use filter-based methods and describing the difference in time resolution between PINE and these methods.

We appreciate this comment and added a part to describe previous filter-based measurements in the sub-Arctic and Arctic in (lines 146-150):

old

new Wex et al. 2019 used filter-based methods to investigate the INP concentration at four different Arctic sites. The measurement duration per filter varied between one to two weeks (Alert, Canada), four days (Ny-Alesund, Svalbard, Norway) and seven days (Utqiaġvik, Alaska, US; Villum Research Station, Greenland, Denmark). The temporal resolution of PINE is higher by a factor of more than 1500, considering a sampling time of seven days. On the contrary, filter-based methods are able to observe lower INP concentrations, especially relevant at higher nucleation temperatures.
* * *
**2 Specific comments about the dataset**

The dataset contains a variable "INP_cn_flush". There is no metadata for this variable and it is not mentioned in the paper. If these are also INP concentrations, they are sometimes 2 orders of magnitude below those in "INP_cn" which was used to generate the figures in this paper, a significant difference. Please could the authors add documentation and then describe this data in the paper.

We are grateful for the careful checking of the dataset and have added some information on this variable in the manuscript (see also answer to reviewer 1, lines 99-102):

old

new The "INP" concentration measured during the flush mode is given in the variable `INP_cn_flush`. This assumes that all aerosols or ice crystals from the walls that are detected during the flush mode above the ice threshold, might appear as an INP during the expansion. This value is typically much lower than the measured INP concentration (`INP_cn`) and is flagged as a "warning" if the condition $\text{INP\_cn\_flush} > 0.5 \times \text{INP\_cn}$ is true.

We have also added the metadata for that variable in the dataset and updated it. Accordingly, we have updated the doi for the dataset in the manuscript (lines 161-162):

old https://doi.org/10.5281/zenodo.13889647, last access: 28022025

new https://doi.org/10.5281/zenodo.16882069, last access: 15082025
* * *
The data is currently in a number of separate files all of different lengths. Sometimes, these clearly represent a single day, but on rare occasions this does not. Is there a particular reason for this?

The data is split into daily files. Since the data is not always in equal time steps this can lead to slight differences in the exact start time of each data file. This is in line with the rest of the data published in the dedicated community Pallas Cloud Experiment - PaCE2022 (`https://zenodo.org/communities/pace2022/`, last access: 28022025).
* * *
L1_data is provided but not raw_data. For clarity, why is the raw data not provided?

The raw_data contains the individual single particle data of the fidas-pine sensor and a number of other data like the valve status. It is of course available upon request, but it is not useful unless the PIA software is used for the analysis and is also much larger in storage compared to the L1_data. An example of a "full" dataset of a PINE can be found here: `https://dx.doi.org/10.35097/sqmdyj7ckbccq9zy`.
* * *
**3 Technical corrections**

Lines 1-2 – It may be useful to rephrase the end of this sentence so the time period is clearer immediately to the reader, as December is typically considered outside autumn. For instance, "…(PaCE) at the Sammaltunturi station between late September 2022 and late December 2022." However, I understand if the authors would rather keep it as is/include the word autumn to make it easier for people to find the measurements using search engines.

We agree that it is useful to specify this more clearly and therefore added it in parentheses (lines 1-2):

old The Portable Ice Nucleation Experiment (PINE) was deployed during the Pallas Cloud Experiment (PaCE) 2022 for a three-month-period at the Sammaltunturi station in autumn 2022.

new The Portable Ice Nucleation Experiment (PINE) was deployed during the Pallas Cloud Experiment (PaCE) 2022 for a three-month-period at the Sammaltunturi station in autumn 2022 (between late September 2022 and late December 2022).
* * *
In my opinion, Section 3 follows a slightly illogical order. I think it would make more sense to first describe the chamber and its general principles of operation, then describe the specifics of the operation in this case.

We agree and therefore have moved the following sentence to the end of Section 3:

old Lines 59-60: During the PaCE-2022 campaign, the PINE was operated between 240 K and 252 K at a temporal resolution of approximately 6 min.

new Lines 78-79: During the PaCE-2022 campaign, the PINE was operated between 240 K and 252 K at a temporal resolution of approximately 6 min.
* * *
Line 74 and corresponding reference – As alluded to later (line 129), v2.0.2 of the PIA software is currently available online. As such, I think this citation should not be "in preparation" but link directly to this version of the code, preferably via permalink to a permanent archive such as Zenodo if possible. P.S. The code is very well-documented, which is excellent!

This reference is for a paper that is currently in preparation for Atmospheric Measurement Techniques, which describes the PIA software in more detail, but we agree that this might be confusing considering the different citations to software and software paper. Therefore, we have updated this to reflect just the software reference (lines 83-84)

old The `L1_data` is created by the custom-made Python software PINE INP Analysis (PIA) v2.0.2 (Büttner et al. 2024, in preparation).

new The `L1_data` is created by the custom-made Python software PINE INP Analysis (PIA) v2.0.2 (Büttner and Fösig 2023).

and also updated the reference link, since the software is also published with a doi in lines 164-165:

old The PIA software is available on a public gitlab instance under `https://codebase.helmholtz.cloud/pine/pia_software`.

new The PIA software is published under (Büttner and Fösig 2025, `https://doi.org/10.5281/zenodo.15592431`).
* * *
**References**

Asmi, E. et al. (2011). "Secondary new particle formation in Northern Finland Pallas site between the years 2000 and 2010". In: *Atmospheric Chemistry and Physics* 11.24, pp. 12959–12972. ISSN: 1680-7324. DOI: `10.5194/acp-11-12959-2011`.

Benz, S. et al. (2005). "T-dependent rate measurements of homogeneous ice nucleation in cloud droplets using a large atmospheric simulation chamber". In: *Journal of Photochemistry and Photobiology A: Chemistry* 176.1–3, pp. 208–217. ISSN: 1010-6030. DOI: `10.1016/j.jphotochem.2005.08.026`.

Büttner, N. and R. Fösig (2023). *PIA Software (v2.0.2)*. en. Zenodo. URL: `https://doi.org/10.5281/zenodo.15592786`.

Büttner, N. and R. Fösig (2025). *PIA Software*. en. Zenodo. URL: `https://doi.org/10.5281/zenodo.15592431`.

Büttner, N. et al. (2024). "Functionality and evaluation of the PINE INP Analysis (PIA) Software". In: in preparation.

Dagsson-Waldhauserova, P., O. Arnalds, and H. Olafsson (2017). "Long-term dust aerosol production from natural sources in Iceland". In: *Journal of the Air & Waste Management Association* 67.2, pp. 173–181. DOI: `10.1080/10962247.2013.805703`.

*Evaluation of measurement data - Guide to the expression of uncertainty in measurement* (2008). Working Group 1 of the Joint Committee for Guides in Metrology.

Gratzl, J. et al. (2025). "Fluorescent aerosol particles in the Finnish sub-Arctic during the Pallas Cloud Experiment 2022 campaign". In: *Earth System Science Data*. in preparation for this SI of ESSD.

Groot Zwaaftink, C. et al. (2017). "Contributions of Icelandic and other high-latitude sources to mineral dust in the Arctic". In: *EGU General Assembly Conference Abstracts*. EGU General Assembly Conference Abstracts, p. 13502.

Meinander, O. et al. (2025). "Dust in the Arctic: a brief review of feedbacks and interactions between climate change, aeolian dust and ecosystems". In: *Frontiers in Environmental Science* 13. ISSN: 2296-665X. DOI: 10.3389/fenvs.2025.1536395.

Möhler, O. et al. (2021). "The Portable Ice Nucleation Experiment (PINE): a new online instrument for laboratory studies and automated long-term field observations of ice-nucleating particles". In: *Atmospheric Measurement Techniques* 14.2, pp. 1143–1166. DOI: 10.5194/amt-14-1143-2021.

Pisso, I. et al. (2019). "The Lagrangian particle dispersion model FLEXPART version 10.4". In: *Geoscientific Model Development* 12.12, pp. 4955–4997. ISSN: 1991-9603. DOI: 10.5194/gmd-12-4955-2019.

Schneider, J. et al. (2020). "The seasonal cycle of ice-nucleating particles linked to the abundance of biogenic aerosol in boreal forests". In: *Atmospheric Chemistry and Physics* 21.5, pp. 3899–3918. DOI: 10.5194/acp-2020-683.

Tobo, Y. et al. (2024). "Surface warming in Svalbard may have led to increases in highly active ice-nucleating particles". In: *Communications Earth & Environment* 5.1. ISSN: 2662-4435. DOI: 10.1038/s43247-024-01677-0.

Vergara-Temprado, J. et al. (2018). "Strong control of Southern Ocean cloud reflectivity by ice-nucleating particles". In: *Proc. Natl. Acad. Sci.* 115.11, pp. 2687–2692. DOI: 10.1073/pnas.1721627115.

Vogel, F. et al. (2024). "Ice-nucleating particles active below -24 °C in a Finnish boreal forest and their relationship to bioaerosols". In: DOI: 10.5194/egusphere-2024-853.

Wex, H. et al. (2019). "Annual variability of ice-nucleating particle concentrations at different Arctic locations". In: *Atmos. Chem. Phys.* 19.7, pp. 5293–5311. ISSN: 1680-7324. DOI: 10.5194/acp-19-5293-2019.